# Physics-aware Spatiotemporal Modules with Auxiliary Tasks for Meta-Learning

## Abstract

Modeling the dynamics of real-world physical systems is critical for spatiotemporal prediction tasks, but challenging when data is limited. The scarcity of real-world data and the difficulty in reproducing the data distribution hinder directly applying meta-learning techniques. Although the knowledge of governing partial differential equations (PDE) of the data can be helpful for the fast adaptation to few observations, it is mostly infeasible to exactly find the equation for observations in real-world physical systems. In this work, we propose a framework, physics-aware meta-learning with auxiliary tasks whose spatial modules incorporate PDE-independent knowledge and temporal modules utilize the generalized features from the spatial modules to be adapted to the limited data, respectively. The framework is inspired by a local conservation law expressed mathematically as a continuity equation and does not require the exact form of governing equation to model the spatiotemporal observations. The proposed method mitigates the need for a large number of real-world tasks for meta-learning by leveraging spatial information in simulated data to meta-initialize the spatial modules. We apply the proposed framework to both synthetic and real-world spatiotemporal prediction tasks and demonstrate its superior performance with limited observations.

## 1 Introduction

Deep learning has recently shown promise to play a major role in devising new solutions to applications with natural phenomena, such as climate change (Manepalli et al., 2019; Drgona et al., 2019), ocean dynamics (Cosne et al., 2019), air quality (Soh et al., 2018; Du et al., 2018; Lin et al., 2018), and so on. Deep learning techniques inherently require a large amount of data for effective representation learning, so their performance is significantly degraded when there are only a limited number of observations. However, in many tasks in physical systems in the real-world we only have access to a limited amount of data. One example is air quality monitoring (Berman, 2017), in which the sensors are irregularly distributed over the space – many sensors are located in urban areas whereas there are much fewer sensors in vast rural areas. Another example is extreme weather modeling and forecasting, i.e., temporally short events (e.g., tropical cyclones (Racah et al., 2017b)) without sufficient observations over time. Moreover, inevitable missing values from sensors (Cao et al., 2018; Tang et al., 2019) further reduce the number of operating sensors and shorten the length of fully-observed sequences. Thus, achieving robust performance from a few spatiotemporal observations in physical systems remains an essential but challenging problem.

Learning on a limited amount of data from physical systems can be considered as a few shot learning. While recently many meta-learning techniques (Schmidhuber, 1987; Andrychowicz et al., 2016; Ravi & Larochelle, 2017; Santoro et al., 2016; Snell et al., 2017; Finn et al., 2017) have been developed to address this few shot learning setting, there are still some challenges for the existing meta-learning methods to be applied in modeling natural phenomena. First, it is not easy to find a set of similar meta-tasks which provide shareable latent representations needed to understand targeted observations. For instance, while image-related tasks (object detection (He et al., 2017) or visual-question-answering tasks (Andreas et al., 2016; Fukui et al., 2016)) can take advantage of an image-feature extractor pre-trained by a large set of images (Deng et al., 2009) and well-designed architecture (Simonyan & Zisserman, 2014; He et al., 2016; Sandler et al., 2018), there is no such large data corpus that is widely applicable for understanding natural phenomena. Second, unlike computer vision or natural language processing tasks where a common object (images or words) is clearly de-

fined, it is not straightforward to find analogous objects in the spatiotemporal data. Finally, exact equations behind natural phenomena are usually unknown, leading to the difficulty in reproducing the similar dataset via simulation. For example, although there have been some works (de Bezenac et al., 2018; Lutter et al., 2019; Greydanus et al., 2019) improving data efficiency via explicitly incorporating PDEs as neural network layers when modeling spatiotemporal dynamics, it is hard to generalize for modeling different or unknown dynamics, which is ubiquitous in real-world scenario.

In this work, we propose physics-aware modules designed for meta-learning to tackle the few shot learning challenges in physical observations. One of fundamental equations in physics describing the transport of physical quantity over space and time is a continuity equation:

$$\frac{\partial \rho}{\partial t} + \nabla \cdot J = \sigma, \tag{1}$$

where $\rho$ is the amount of the target quantity ($u$) per unit volume, $J$ is the flux of the quantity, and $\sigma$ is a source or sink, respectively. This fundamental equation can be used to derive more specific transport equations such as the convection-diffusion equation, Navier-Stokes equations, and Boltzmann transport equation. Thus, the continuity equation is the starting point to model spatiotemporal (conservative) observations which are accessible from sensors. Based on the form of $\rho$ and $J$ with respect to a particular quantity $u$, Eq. 1 can be generalized as:

$$\frac{\partial u}{\partial t} = F(\nabla u, \nabla^2 u, \dots), \tag{2}$$

where the function $F(\cdot)$ describes how the target $u$ is changed over time from its spatial derivatives. Inspired by the form of Eq. 2, we propose two modules: spatial derivative modules (SDM) and time derivative modules (TDM). Since the spatial derivatives such as $\nabla$, $\nabla\cdot$, and $\nabla^2$ are commonly used across different PDEs, the spatial modules are PDE-independent and they can be meta-initialized from synthetic data. Then, the PDE-specific temporal module is trained to learn the unknown function $F(\cdot)$ from few observations in the real-world physical systems.

This approach can effectively leverage a large amount of simulated data to train the spatial modules as the modules are PDE-independent and thus mitigating the need for a large amount of real-world tasks to extract shareable features. In addition, since the spatial modules are universally used in physics equations, the representations from the modules can be conveniently integrated with data-driven models for modeling natural phenomena. Based on the modularized PDEs, we introduce a novel approach that marries physics knowledge in spatiotemporal prediction tasks with meta-learning by providing shareable modules across spatiotemporal observations in the real-world.

Our contributions are summarized below:

- **Modularized PDEs and auxiliary tasks:** Inspired by forms of PDEs in physics, we decompose PDEs into shareable (spatial) and adaptation (temporal) parts. The shareable one is PDE-independent and specified by auxiliary tasks: *supervision of spatial derivatives*.

- **Physics-aware meta-learning:** We provide a framework for physcis-aware meta-learning, which consists of PDE-independent/-specific modules. The framework is flexible to be applied to the modeling of different or unknown dynamics.

- **Synthetic data for shareable modules:** We extract shareable parameters in the spatial modules from synthetic data, which can be generated from different dynamics easily.

## 2 MODULARIZED PDEs AND META-LEARNING

In this section, we describe how the physics equations for conserved quantities are decomposable into two parts and how the meta-learning approach tackles the task by utilizing synthetic data when the data are limited.

### 2.1 DECOMPOSABILITY OF VARIANTS OF A CONTINUITY EQUATION

In physics, a continuity equation (Eq. 1) describes how a locally conserved quantity such as temperature, fluid density, heat, and energy is transported across space and time. This equation underlies

many specific equations such as the convection-diffusion equation and Navier-Stokes equations:

$$\dot{u} = \nabla \cdot (D\nabla u) - \nabla \cdot (\boldsymbol{v}u) + R, \qquad \text{(Convection-Diffusion eqn.)}$$

$$\dot{\boldsymbol{u}} = -(\boldsymbol{u} \cdot \nabla)\boldsymbol{u} + \nu\nabla^2\boldsymbol{u} - \nabla\omega + \boldsymbol{g}. \qquad \text{(Incompressible Navier-Stokes eqn.)}$$

where the scalar $u$ and vector field $\boldsymbol{u}$ are the variables of interest (e.g., temperature, flow velocity, etc.). A dot over a variable is time derivative. The common feature in these equations is that the forms of equations can be digested as (Bar-Sinai et al., 2019; Zhuang et al., 2020):

$$\dot{u} = F(u_x, u_y, u_{xx}, u_{yy}, \dots), \tag{3}$$

where the right-hand side denotes a function of spatial derivatives. As the time derivative can be seen as a Euler discretization (Chen et al., 2018), it is notable that the next state is a function of the current state and spatial derivatives. Thus, knowing spatial derivatives at time $t$ is a key step for spatiotemporal prediction at time $t + 1$ for locally conserved quantities. According to Eq. 3, the spatial derivatives are universally used in variants of Eq. 1 and only the updating function $F(\cdot)$ is specifically defined for a particular equation. This property implies that PDEs for physical quantities can be decomposable into two modules: spatial and temporal derivative modules.

## 2.2 SPATIAL DERIVATIVE MODULES: PDE-INDEPENDENT MODULES

Finite difference method (FDM) is widely used to discretize a $d$-order derivative as a linear combination of function values on a $n$-point stencil.

$$\frac{\partial^d u}{\partial x^d} \approx \sum_{i=1}^{n} \alpha_i u(x_i), \tag{4}$$

where $n > d$. According to FDM, it is independent for a form of PDE to compute spatial derivatives, which are input components of $F(\cdot)$ in Eq. 3. Thus, we can modularize spatial derivatives as PDE-independent modules. The modules that can be learnable as a data-driven manner to infer the coefficients ($\alpha_i$) have been proposed recently (Bar-Sinai et al., 2019; Seo et al., 2020). The data-driven coefficients are particularly useful when the discretization in the $n$-point stencil is irregular and low-resolution where the fixed coefficients cause substantial numerical errors.

## 2.3 TIME DERIVATIVE MODULE: PDE-SPECIFIC MODULE

Once upto $d$-order derivatives are modularized by learnable parameters, the approximated spatial derivatives from the spatial modules are fed into an additional module to learn the function $F(\cdot)$ in Eq. 3. This module is PDE-specific as the function $F$ describes how the spatiotemporal observations change. Since the exact form of a ground truth PDE is not given, the time derivative module is data-driven and will be adapted to observations instead.

## 2.4 META-LEARNING WITH PDE-INDEPENDENT/-SPECIFIC MODULES

Recently, Raghu et al. (2019) investigate the effectiveness of model agnostic meta-learning (MAML, Finn et al. (2017)) and it is found that the outer loop of MAML is more likely to learn parameters for reusable features rather than rapid adaptation. The finding that feature reuse is the predominant reason for efficient learning of MAML allows us to use additional information which is beneficial for learning better representations. Previously, the objective in meta-training has been considered to be matched with one in meta-test as the purpose of meta-learning is to learn good initial parameters applicable across similar tasks (e.g., image classification to image classification). We are now able to incorporate auxiliary tasks under a meta-learning setting to reinforce reusable features for a main task. As described in Sec. 2.1, the spatial modules are reusable across different observations, and thus, we can meta-initialize the spatial modules first with spatial derivatives provided by synthetic datasets. Then, we can integrate the spatial modules with the task-specific temporal module during meta-test to help adaptation of TDM on few observations. Since the spatial modules are trained by readily available synthetic datasets, a large number of similar tasks for meta-training is not required.

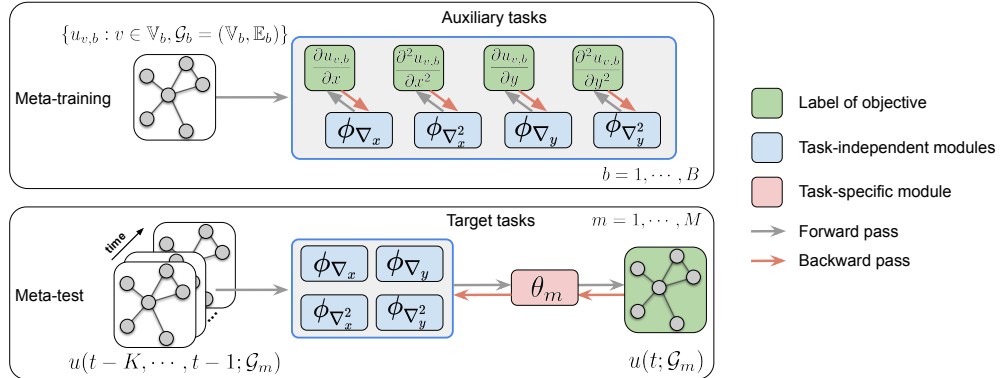

Figure 1: Schematic overview of the physics-aware meta-learning (PiMetaL).

# 3 PHYSICS-AWARE META-LEARNING WITH AUXILIARY TASKS

In this section, we develop a physics-aware meta-learning framework for the modularized PDEs. Fig. 1 describes the proposed framework and its computational process.

## 3.1 SPATIAL DERIVATIVE MODULE

---
**Algorithm 1** Spatial derivative module (SDM)
---
**Input**: Graph signals $u_i$ and edge features $\boldsymbol{e}_{ij} = \boldsymbol{x}_j - \boldsymbol{x}_i$ on $\mathcal{G}$ where $\boldsymbol{x}_i$ is a coordinate of node $i$.
**Output**: Spatial derivatives $\{\hat{u}_{k,i} \mid i \in \mathbb{V} \text{ and } k \in \mathbb{K}\}$ where $\mathbb{K} = \{\nabla_x, \nabla_y, \nabla_x^2, \nabla_y^2\}$.
**Require**: Spatial derivative modules $\{\phi_k \mid k \in \mathbb{K}\}$

  1: **for** $k \in \mathbb{K}$ **do**
  2:     $\{a_{k,i}, b_{k,(i,j)} \mid (i,j) \in \mathbb{E} \text{ and } k \in \mathbb{K}\} = \phi_k(\{u\}, \{\boldsymbol{e}\}, \mathcal{G})$
  3:     **for** $i \in \mathbb{V}$ **do**
  4:         $\hat{u}_{k,i} = a_{k,i}u_i + \sum_{(j,i)\in\mathbb{E}} b_{k,(j,i)}u_j$
  5:     **end for**
  6: **end for**

---

As we focus on the modeling and prediction of sensor-based observations, where the available data points are inherently on a spatially sparse irregular grid, we use graph networks for each module $\phi_k$ to learn the finite difference coefficients (Bar-Sinai et al., 2019). Given a graph $\mathcal{G} = (\mathbb{V}, \mathbb{E})$ where $\mathbb{V} = \{1, \ldots, N\}$ and $\mathbb{E} = \{(i,j) : i, j \in \mathbb{V}\}$, a node $i$ denotes a physical location $\boldsymbol{x}_i = (x_i, y_i)$ where a function value $u_i = u(x_i, y_i)$ is observed. Then, the graph signals with positional relative displacement as edge features are fed into the spatial modules to approximate spatial derivatives by Alg. 1. The coefficients $(a_i, b_{(i,j)})$ on each node $i$ and edge $(i,j)$ are output of $\phi$ and they are linearly combined with the function values $u_i$ and $u_j$. $\mathbb{K}$ denotes a set of finite difference operators. For example, if we set $\mathbb{K} = \{\nabla_x, \nabla_y, \nabla_x^2, \nabla_y^2\}$, we have 4 modules which approximate first/second order of spatial derivatives in 2-dimension, respectively.

## 3.2 TIME DERIVATIVE MODULE

Once spatial derivatives are approximated, another learnable module is required to combine them for a target task. The form of line 2 in Alg. 2 comes from Eq. 3 and TDM is adapted to learn the unknown function $F(\cdot)$ in the equation. As our target task is the regression of graph signals, we use a recurrent graph network for TDM.

---
**Algorithm 2** Time derivative module (TDM)
---
**Input**: Graph signals $u$ and approximated spatial derivatives $\hat{u}_k$ where $k \in \mathbb{K}$ on $\mathcal{G}$. Time interval $\Delta t$
**Output**: Prediction of signals at next time step $\hat{u}(t)$
**Require**: Time derivative module

  1: $\hat{u}_t = \text{TDM}(\{u_i, \hat{u}_{k,i} \mid i \in \mathbb{V} \text{ and } k \in \mathbb{K}\})$
  2: $\hat{u}(t) = u(t-1) + \hat{u}_{t-1} \cdot \Delta t$

---

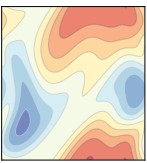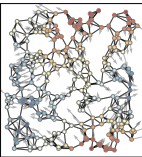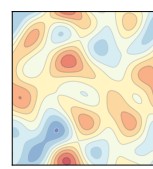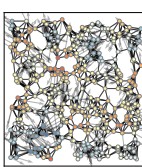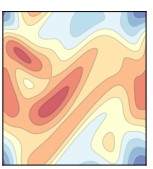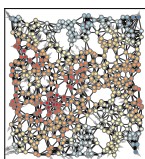

Figure 2: Examples of generated spatial function values and graph signals. Node and edge features (function value and relative displacement, respectively) are used to approximate spatial derivatives (arrows). We can adjust the number of nodes (spatial resolution), the number of edges (discretization), and the degree of fluctuation (scale of derivatives) to differentiate meta-train tasks.

## 3.3 META-LEARNING WITH AUXILIARY OBJECTIVE

As discussed in Sec. 2.1, it is important to know spatial derivatives at time $t$ to predict next signals at time $t + 1$ for locally conserved physical quantities, however, it is impractical to access the spatial derivatives in the sensor-based observations as they are highly discretized over space. In this section, we propose a physics-aware meta-learning framework to meta-initialize a spatial module by leveraging synthetic dataset with auxiliary tasks to provide reusable features for the main tasks: prediction spatiotemporal observations in the real-world.

The meta-initialization with the auxiliary tasks from synthetic datasets is particularly important. First, the spatial modules can be universal feature extractors for modeling observations following unknown physics-based PDEs. Unlike other domains such as computer vision, it has been considered that there is no particular shareable architecture for learning spatiotemporal dynamics from physical systems. We propose that the PDE-independent spatial modules can be applicable as feature extractors across different dynamics as long as the dynamics follow a local form of conservation laws. Second, we can utilize synthetic data to meta-train the spatial modules as they are PDE-agnostic. This property allows us to utilize a large amount of synthetic datasets which are readily generated by numerical methods regardless of the exact form of PDE for targeted observations. Finally, we can provide a stronger inductive bias which is beneficial for modeling real-world observations but not available in the observations explicitly.

---

**Algorithm 3** Meta-initialization with auxiliary tasks: Supervision of spatial derivatives

---

**Input**: A set of meta-train task datasets $\mathcal{D} = \{\mathcal{D}_1, \ldots, \mathcal{D}_B\}$ where $\mathcal{D}_b = (\mathcal{D}_b^{tr}, \mathcal{D}_b^{te})$.
$\mathcal{D}_b = \{(u_i^b, e_{ij}^b, y_i^{(a_1,b)}, \ldots, y_i^{(a_K,b)}) : i \in \mathbb{V}_b, (i,j) \in \mathbb{E}_b\}$ where $y_i^{(a_k,\cdot)}$ is an $k$-th auxiliary task label for the $i$-th node, given node/edge feature $u^b$ and $e^b$, respectively. Learning rate $\alpha$ and $\beta$.
**Output**: Meta-initialized spatial modules $\Phi$.

1: Initialize auxiliary modules $\Phi = (\phi_1, \ldots, \phi_K)$
2: **while** not converged **do**
3:   **for** $\mathcal{D}_b$ in $\mathcal{D}$ **do**
4:     $\Phi_b' = \Phi - \alpha \nabla_\Phi \sum_{k=1}^K \mathcal{L}_k^{aux}(\mathcal{D}_b^{tr}; \phi_k)$
5:   **end for**
6:   $\Phi \leftarrow \Phi - \beta \nabla_\Phi \sum_{b=1}^B \sum_{k=1}^K \mathcal{L}_k^{aux}(\mathcal{D}_b^{te}; \phi_{b,k}')$
7: **end while**

---

Alg. 3 describes how the spatial modules are meta-initialized by MAML under the supervision of $K$ different spatial derivatives. First, we generate values and spatial derivatives on a 2D regular grid from an analytical function. Then, we sample a finite number of points from the regular grid to represent discretized nodes and build a graph from the sampled nodes. Each graph signal and its discretization becomes input feature of a meta-train task and corresponding spatial derivatives are the auxiliary task labels. Fig. 2 visualizes graph signals and spatial derivatives for meta-initialization.

Once the spatial modules are initialized throughout meta-training, we reuse the modules for meta-test where the temporal module (the head of the network) are adapted on few observations from real-world sensors (Alg. 4). Although the standard MAML updates the body of the network (the spatial modules) as well, we only adapt the head layer ($\theta$) as like almost-no-inner-loop method

---

**Algorithm 4** Adaptation on meta-test tasks

---

**Input**: A set of meta-test task datasets $\mathcal{D} = \{\mathcal{D}_1, \ldots, \mathcal{D}_M\}$ where $\mathcal{D}_b = (\mathcal{D}_m^{tr}, \mathcal{D}_m^{te})$.
Meta-initialized SDM ($\Phi$). Learning rate $\alpha$.
**Output**: Adapted TDM $\theta'_m$ on the $m$-th task.

1:  Initialize temporal modules $(\theta_1, \ldots, \theta_M)$
2:  **for** $\mathcal{D}_m$ in $\mathcal{D}$ **do**
3:      $\theta'_m = \theta_m - \alpha \nabla_{\theta_m} \mathcal{L}(\mathcal{D}_m^{tr}; \Phi, \theta_m)$
4:  **end for**

---

in Raghu et al. (2019). The task at test time is graph signal prediction and the temporal modules ($\theta$) are adapted by a regression loss function $\mathcal{L} = \sum_{t=1}^{T} ||u(t) - \hat{u}(t)||^2$ on length $T$ sequence ($\mathcal{D}_m^{tr}$) and evaluated on held-out ($t > T$) sequence ($\mathcal{D}_m^{te}$) with the adapted parameters.

## 4  SPATIAL DERIVATIVE MODULES: REUSABLE MODULES

We have claimed that the spatial modules provide reusable features associated with spatial derivatives such as $\nabla_x u, \nabla_y u$, and $\nabla_x^2 u$ across different dynamics or PDEs. While it has been shown that the data-driven approximation of spatial derivatives is more precise than that of finite difference method (Seo et al., 2020; Bar-Sinai et al., 2019), it is not guaranteed that the modules effectively provide transferrable parameters for different spatial resolution, discretization, and fluctuation of function values. We explore whether the proposed spatial derivative modules based on graph networks can be used as a feature provider for different spatial functions and discretization.

We perform two sets of experiments: evaluate few-shot learning performance (1) when SDM is trained from scratch; (2) when SDM is meta-initialized. Fig. 2 shows how the graph signal and its discretization is changed over the different settings. If the number of nodes is large, it can provide spatially high-resolution and thus, the spatial derivatives can be more precisely ap-

Table 1: Parameters for synthetic dataset.

|  | Meta-train | Meta-test |
|---|---|---|
| # nodes ($N$) | $\{256, 625\}$ | $\{450, 800\}$ |
| # edges per a node ($E$) | $\{4, 8\}$ | $\{3, 6, 10\}$ |
| Initial frequency ($F$) | $\{2, 5\}$ | $\{3, 7\}$ |

proximated. Table 1 shows the parameters we used to generate synthetic datasets. Note that meta-test data is designed to evaluate interpolation/extrapolation properties. Initial frequency decides the degree of fluctuation (In Fig. 2, the middle one has higher $F$ than that of the left one.). For each parameter combination, we generate 100 different snapshots from the following form in Long et al. (2017):

$$u_i = \sum_{|k|,|l| \leq F} \lambda_{k,l} \cos(kx_i + ly_i) + \gamma_{k,l} \sin(kx_i + ly_i), \ \lambda_{k,l}, \gamma_{k,l} \sim \mathcal{N}(0, 0.02), \tag{5}$$

where the index $i$ denotes the $i$-th node whose coordinate is $(x_i, y_i)$ in the 2D space ($[0, 2\pi] \times [0, 2\pi]$) and $k, l$ are randomly sampled integers. From the synthetic data, the first- and second-order derivatives are analytically given and SDM is trained to approximate them.

The prediction results for spatial derivatives are shown in Table 2. The results show that the proposed module (SDM) is efficiently adaptable to different configuration on few samples from meta-initialized parameters compared to learning from scratch. The finding implies that the parameters for spatial derivatives can be generally applicable across different spatial resolution, discretization, and function fluctuation.

## 5  EXPERIMENTAL EVALUATION

### 5.1  PRELIMINARY: WHICH SYNTHETIC DYNAMICS NEED TO BE GENERATED?

While Table 2 demonstrates that the PDE-independent representations are reusable across different configurations, it is still an open question: which topological configuration needs to be used to construct the synthetic dynamics? According to Table 2, the most important factor affecting error is

Table 2: Prediction error (MAE) of the first (top) and second (bottom) order spatial derivatives.

| $(N, E, F)$ | (450,3,3) | (450,3,7) | (450,6,3) | (450,6,7) | (450,10,3) | (450,10,7) |
|---|---|---|---|---|---|---|
| SDM (from scratch) | 1.337±0.044 
 7.278±0.225 | 7.111±0.148 
 51.544±0.148 | 1.152±0.043 
 5.997±0.083 | 7.206±0.180 
 47.527±0.768 | 1.112±0.036 
 5.353±0.193 | 7.529±0.241 
 47.356±0.560 |
| SDM (pretrained) | **1.075±0.005** 
 **6.482±0.207** | **5.528±0.010** 
 **46.254±0.262** | **0.836±0.002** 
 **5.251±0.245** | **5.354±0.001** 
 **42.243±0.420** | **0.782±0.006** 
 **4.728±0.244** | **5.550±0.012** 
 **42.754±0.442** |
| $(N, E, F)$ | (800,3,3) | (800,3,7) | (800,6,3) | (800,6,7) | (800,10,3) | (800,10,7) |
| SDM (from scratch) | 1.022±0.030 
 7.196±0.159 | 5.699±0.242 
 49.602±0.715 | 0.789±0.021 
 5.386±0.136 | 5.179±0.069 
 42.509±1.080 | 0.718±0.010 
 4.536±0.204 | 5.517±0.110 
 39.642±1.173 |
| SDM (pretrained) | **0.927±0.006** 
 **6.553±0.193** | **4.415±0.011** 
 **44.591±0.002** | **0.656±0.008** 
 **4.960±0.266** | **3.977±0.025** 
 **37.629±0.760** | **0.570±0.006** 
 **4.213±0.275** | **4.107±0.019** 
 **35.849±0.947** |

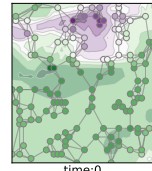 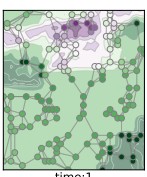 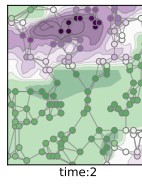 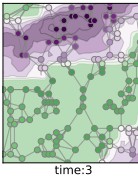 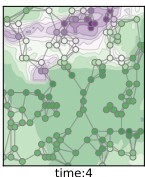

time:0     time:1     time:2     time:3     time:4

Figure 3: Visualization of the first 5 frames of one extended sequence in the extreme weather dataset. Dots represent the sampled points. Greenish (purplish) area is higher (lower) surface temperature.

an initial frequency ($F$), which determines min/max scales and fluctuation of function values, and it implies that the synthetic dynamics should be similarly scaled to a target dynamics. We use the same topological configuration in Table 1 to generate synthetic datasets for a task in Section 5.2 and adapted configuration for a task in Section 5.3. We describe more details in Appendix B.

## 5.2 MULTI-STEP GRAPH SIGNAL GENERATION

**Task**: We adopt a set of multi-step spatiotemporal sequence generation tasks to evaluate our proposed framework. In each task, the data is a sequence of $L$ frames, where each frame is a set of observations on $N$ nodes in space. Then, we train an auto-regressive model with the first $T$ frames ($T$-shot) and generate the following $L - T$ frames repeatedly from a given initial input ($T$-th frame) to evaluate its performance.

**Datasets**: For all experiments, we generate meta-train tasks with the parameters described in Table 1 and the target observations are 2 real-world datasets: (1) **AQI-CO**: national air quality index (AQI) observations (Berman, 2017); (2) **ExtremeWeather**: the extreme weather dataset (Racah et al., 2017b). For the AQI-CO dataset, we construct 12 meta-test tasks with the carbon monoxide (CO) ppm records from the first week of each month in 2015 at land-based stations. For the extreme weather dataset, we select the top-10 extreme weather events with the longest lasting time from the year 1984 and construct a meta-test task from each event with the

Table 3: Multi-step prediction results (MSE) and standard deviations on the two real-world datasets.

| $T$-shot | Method | AQI-CO | ExtremeWeather |
|---|---|---|---|
| 5-shot | FDM+RGN (scratch) | 0.0291±0.0039 | 0.9883±0.5567 |
| | PA-DGN (scratch) | 0.0363±0.0090 | 0.9653±0.1384 |
| | PiMetaL (meta-init) | **0.0253±0.0055** | **0.9167±0.0746** |
| 7-shot | FDM+RGN (scratch) | 0.0258±0.0023 | 0.7626±0.0602 |
| | PA-DGN (scratch) | 0.0225±0.0018 | 0.7478±0.0199 |
| | PiMetaL (meta-init) | **0.0182±0.0019** | **0.7274±0.0089** |
| 10-shot | FDM+RGN (scratch) | 0.0213±0.0013 | 0.7090±0.0030 |
| | PA-DGN (scratch) | 0.0146±0.0005 | 0.4156±0.0145 |
| | PiMetaL (meta-init) | **0.0115±0.0004** | **0.4066±0.0247** |

Table 4: Graph signal regression results (MSE, $10^{-3}$) and standard deviations on the two regions of weather stations.

| $T$-shot (Region) | GCN | GAT | GraphSAGE | GN | PA-DGN | PiMetaL |
|---|---|---|---|---|---|---|
| 5-shot (USA) | 2.742±0.120 | 2.549±0.115 | 2.128±0.146 | 2.252±0.131 | 1.950±0.152 | **1.794±0.130** |
| 10-shot (USA) | 2.371±0.095 | 2.178±0.066 | 1.848±0.206 | 1.949±0.115 | 1.687±0.104 | **1.567±0.103** |
| 5-shot (EU) | 1.218±0.218 | 1.161±0.234 | 1.165±0.248 | 1.181±0.210 | 0.914±0.167 | **0.781±0.019** |
| 10-shot (EU) | 1.186±0.076 | 1.142±0.070 | 1.044±0.210 | 1.116±0.147 | 0.831±0.058 | **0.773±0.014** |

observed surface temperatures at randomly sampled locations. Since each event lasts fewer than 20 frames, each task has a very limited amount of available data. In both datasets, graph signals are univariate. Note that all quantities have fluidic properties such as diffusive and convection. Fig. 3 shows the spatiotemporal dynamics of the extreme weather observations and sampled points. More details are in the supplementary material.

**Baselines**: We evaluate the performance of a physics-aware architecture (**PA-DGN**) (Seo et al., 2020), which also consists of spatial derivative modules and recurrent graph networks (RGN), to see how the additional spatial information affects prediction performance for same architecture. Note that PA-DGN has same modules in PiMetaL and the difference is that PiMetaL utilizes meta-initialized spatial modules and PA-DGN is randomly initialized for learning from scratch on meta-test tasks. Additionally, the spatial modules in PA-DGN is replaced by finite difference method (**FDM+RGN**) to see if the numerical method provides better PDE-agnostic representations. The baselines and PiMetaL are trained on the meta-test support set only to demonstrate how the additional spatial information is beneficial for few-shot learning tasks.

**Discussion**: Table 3 shows the multi-step prediction performance of our proposed framework against the baselines on real-world datasets. Overall, PA-DGN and PiMetaL show similar trend such that the prediction error is decreased as longer series are available for few-shot adaptation. There are two important findings: first, with the similar expressive power in terms of the number of learnable parameters, the meta-initialized spatial modules provide high quality representations which are easily adaptable across different spatiotemporal dynamics in the real-world. This performance gap demonstrates that we can get a stronger inductive bias from synthetic datasets without knowing PDE-specific information. Second, the contribution of the meta-initialization is more significant when the length of available sequence is shorter ($T = 5$) and this demonstrates when the meta-initialization is particularly effective. Finally, the finite difference method provides proxies of exact spatial derivatives and the representations are useful particularly when $T = 5$ but its performance is rapidly saturated and it comes from the gap between the learnable spatial modules and fixed numerical coefficients. The results provide a new point of view on how to utilize synthetic or simulated datasets to handle challenges caused by limited datasets.

## 5.3 GRAPH SIGNAL REGRESSION

**Task, datasets, and baselines**: Defferrard et al. (2019) conducted a graph signal regression task: predict the temperature $x_t$ from the temperature on the previous 5 days ($x_{t-5} : x_{t-1}$). We split the **GHCN** dataset[1] spatially into two regions: (1) the USA (1,705 stations) and (2) Europe (EU) (703 stations) where there are many weather stations full functioning. In this task, the number of shots is defined as the number of input and output pairs to train a model. As the input length is fixed, more variants of graph neural networks are considered as baselines. We concatenate the 5-step signals and feed it into Graph convolutional networks (**GCN**) (Kipf & Welling, 2017), Graph attention networks (**GAT**) (Veličković et al., 2018), **GraphSAGE** (Hamilton et al., 2017), and Graph networks (**GN**) (Battaglia et al., 2018) to predict next signals across all nodes.

**Discussion**: Table 4 shows the results of the graph signal regression task across different baselines and the proposed method. There are two patterns in the results. First, although in general we observe an improvement in performance for all methods when we move from the 5-shot setting to the 10-shot setting, PiMetaL's performance yields the smallest error. Second, for the EU dataset, while 5-shot seems enough to achieve stable performance, it demonstrates that the PDE-independent

---

[1]Global Historical Climatology Network (GHCN) provided by National Oceanic and Atmospheric Administration (NOAA). https://www.ncdc.noaa.gov/ghcn-daily-description

representations make the regression error converge to a lower level. Overall, the experimental results prove that the learned spatial representations from simulated dynamics are beneficial for learning on limited data.

## 6  RELATED WORK

**Physics-informed learning**   Since physics-informed neural networks are introduced in Raissi et al. (2019), which find that a solution of a PDE can be discovered by neural networks, physical knowledge has been used as an inductive bias for deep neural networks. Advection-diffusion equation is incorporated with deep neural networks for sea-surface temperature dynamics (de Bezenac et al., 2018). Lutter et al. (2019); Greydanus et al. (2019) show that Lagrangian/Hamiltonian mechanics can be imposed to learn the equations of motion of a mechanical system and Seo & Liu (2019) regularizes a graph neural network with a specific physics equation. Rather than using explicitly given equations, physics-inspired inductive bias is also used for reasoning dynamics of discrete objects (Battaglia et al., 2016; Chang et al., 2016) and continuous quantities (Seo et al., 2020). Long et al. (2017; 2019) propose a numeric-symbolic hybrid deep neural network designed to discover PDEs from observed dynamic data. While there are many physics-involved works, to the best of our knowledge, we are the first to provide a framework to use the physics-inspired inductive bias under the meta-learning settings to tackle the limited data issue which is pretty common for real-world data such as extreme weather events (Racah et al., 2017b).

**Meta-learning**   The aim of meta-learning is to enable learning parameters which can be used for new tasks unknown at the time of learning, leading to agile models which adapt to a new task utilizing only a few samples (Schmidhuber, 1987; Naik & Mammone, 1992; Thrun & Pratt, 1998). Based on how the knowledge from the related tasks is used, meta-learning methods have been classified as optimization-based (Andrychowicz et al., 2016; Ravi & Larochelle, 2017; Duan et al., 2017; Finn et al., 2017; Nichol et al., 2018; Antoniou et al., 2018; Rusu et al., 2018; Grant et al., 2018), model-based (Santoro et al., 2016; Munkhdalai & Yu, 2017; Duan et al., 2017; Mishra et al., 2018), and metric-based (Koch et al., 2015; Vinyals et al., 2016; Snell et al., 2017). Recently, another branch of meta-learning has been introduced to more focus on finding a set of reusable modules as components of a solution to a new task. Alet et al. (2018; 2019) provide a framework, structured modular meta-learning, where a finite number of modules are introduced as task-independent modules and an optimal structure combining the modules is found from a limited number of data. Chen et al. (2019) introduces techniques to automatically discover task-independent/dependent modules based on Bayesian shrinkage to find more adaptable modules. To our knowledge, none of the above works provide a solution to use meta-learning for modeling physics-related spatiotemporal dynamics where it is hard to generate enough tasks for meta-initialization.

## 7  CONCLUSION

In this paper, we propose a framework for physics-aware meta-learning with auxiliary tasks. By incorporating PDE-independent/-invariant knowledge (spatial derivatives) from simulated data, the framework provide reusable features to meta-test tasks with a limited amount of data. Experiments show that auxiliary tasks and physics-aware meta-learning help construct reusable modules that improve the performance of spatiotemporal predictions in real-world tasks where data is limited. Although introducing auxiliary tasks based on synthetic datasets improves the prediction performance, they need to be chosen and constructed manually and intuitively. Designing and identifying the most useful auxiliary tasks and data will be the focus of our future work.

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

# A  TASK 1: MULTI-STEP GRAPH SIGNAL GENERATION

## A.1  META-TRAIN

**Data**: For all experiments, we generate the data for meta-train tasks from a sum of sinusoidal functions with different spatial frequencies (Eq. 6).

$$u(x, y) = \sum_{|k|,|l| \leq F} \lambda_{k,l} \cos(kx + ly) + \gamma_{k,l} \sin(kx + ly), \ \lambda_{k,l}, \gamma_{k,l} \sim \mathcal{N}(0, 0.02), \tag{6}$$

where $(x, y)$ in the 2D space $([0, 2\pi] \times [0, 2\pi])$ and $k, l$ are randomly sampled integers. Once the spatially continuous function values are generated, we uniformly sample different number of locations from all grid points as observed nodes to simulate the case where the observations are irregularly distributed in space. We then construct a $k$-Nearest Neighbor graph based on the Euclidean distance as the input of graph neural networks. The combination of parameters to generate the synthetic dataset is given in Table 5. We construct 100 snapshots per a combination of the parameters $(N, E, F)$ using a unique random seed. 75 snapshots per each combination are used for $\mathcal{D}^{tr}$ and 25 snapshots are for $\mathcal{D}^{te}$.

Table 5: Parameters for synthetic dataset

|  | Meta-train | Meta-test |
|---|---|---|
| # nodes ($N$) | {256, 625} | {450, 800} |
| # edges per a node ($E$) | {4, 8} | {3, 6, 10} |
| Initial frequency ($F$) | {2, 5} | {3, 7} |

**Tasks**: For each node, we have the first and second order derivatives. We meta-train the spatial derivative modules (Sec. 3.1) to predict the spatial derivatives by feeding node and edge features (function value at a node and relative displacement, respectively) as input.

## A.2  META-TEST

### A.2.1  SYNTHETIC

**Data**: We generate the synthetic meta-test data from Eq. 6 but set different parameters to simulate the realistic scenario where meta-train tasks and meta-test tasks do not share the same distribution.

**Tasks**: We reuse the spatial modules in A.1 to evaluate how the meta-initialized parameters are easily adaptable to unseen graph signals with different spatial resolution, discretization, and the degree of function fluctuation. We use 15 snapshots for the adaptation in meta-test and 75 snapshots are used to evaluate the proposed model.

### A.2.2  REAL-WORLD DATASET

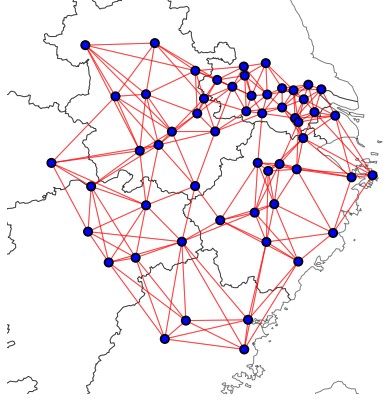

Figure 4: Sensor locations in the AQI-CO dataset. We show sensors as blue nodes and edges of $k$-NN graphs as red lines. Borders of provinces are shown in grey.

**Data**:

**AQI-CO** (Berman, 2017): There are multiple pollutants in the dataset and we choose carbon monoxide (CO) ppm as a target pollutant in this paper. We select sensors located in between latitude (26, 33) and longitude (115,125) (East region of China). In this region, we sample multiple multivariate time series whose length should be larger than 12 steps (12 hours) for multiple meta-tasks. There are around 60 working sensors and the exact number of the working sensors is varying over different tasks. Fig. 4 shows the locations of selected AQI sensors.

**ExtremeWeather**: We select the data in the year 1984 from the extreme weather dataset in (Racah et al., 2017a). The data is an array of shape (1460, 16, 768, 1152), containing 1460 frames (4 per day, 365 days in the year). 16 channels in each frame correspond to 16 spatiotemporal variables. Each channel has a size of $768 \times 1152$ corresponding to one measurement per 25 square km on earth. For each frame, the dataset provides fewer than or equal to 15 bounding boxes, each of which labels the region affected by an extreme weather event and one of the four types of the extreme weather: (1) tropical depression, (2) tropical cyclone, (3) extratropical cyclone, (4) atmospheric river. In the single feature setting, we only utilize the channel of surface temperature (TS).

**Tasks**:

**AQI-CO**: We select the first sequence of carbon monoxide (CO) ppm records from each month in the year 2015 at land-based stations, and set up the meta-test task on each sequence as the prediction of CO ppm. We construct a 6-NN graph based on the geodesic distances among stations.

**ExtremeWeather**: First, we aggregate all bounding boxes into multiple sequences. In each sequence, all bounding boxes (1) are in consecutive time steps, (2) are affected by the same type of extreme weather, and (3) have an intersection over union (IoU) ratio above 0.25 with the first bounding box in the sequence. Then we select the top-10 longest sequences. For each sequence, we consider its first bounding box $A$ as the region affected by an extreme weather event, and extend it to a new sequence of 20 frames by cropping and appending the same region $A$ from successive frames. For each region we uniformly sample 10% of available pixels as observed nodes to simulate irregularly spaced weather stations and build a 4-NN graph based on the Euclidean distance. Fig. 3 visualizes the first 5 frames of one extended sequence. In the single feature experiment, we set up a meta-test task on each extended sequence as the prediction of the surface temperature (TS) on all observed nodes with the initial TS given only.

## A.3 EXPERIMENTAL DETAILS

### A.3.1 BASELINES

**PA-DGN (train from scratch)** (Seo et al., 2020): For each meta-test task, initialize one PA-DGN model randomly and train it on the single task. The spatial derivative layer uses a message passing neural network (MPNN) with 2 GN blocks using 2-layer MLPs as update functions. The forward network part uses a recurrent graph neural network with 2 recurrent GN blocks using 2-layer GRU cells as update functions. We set its hidden dimension to 64, in which case PA-DGN has a similar number of parameters with RGN. The PA-DGN model has 384,653 learnable parameters.

### A.3.2 OURS

**PiMetaL**: Meta-train the spatial derivative modules (SDM) with our proposed Alg. 3 on the meta-train tasks generated in A.1. Then for each meta-test task, initialize one time derivative module (TDM) randomly and output of SDM is fed into the TDM to train it on the single task. The architecture for SDM and TDM are identical for the spatial derivative layer and the recurrent graph network in PA-DGN, respectively.

### A.3.3 TRAINING SETTINGS

**Training hyperparameters**: For all meta-train and meta-test tasks, we use the Adam optimizer with the learning rate 1e-3. In each training epoch, we sample 1 task from all available tasks.

**Environments**: All experiments are implemented with Python3.6 and PyTorch 1.3.0, and are conducted with NVIDIA GTX 1080 Ti GPUs.

**Runtime**: The baselines RGN (train from scratch) and PA-DGN (train from scratch) will finish in 30 minutes. All other baselines will finish the meta-train stage in 4 hours and the meta-test stage in 2 hours. The runtime is measured in environments described above.

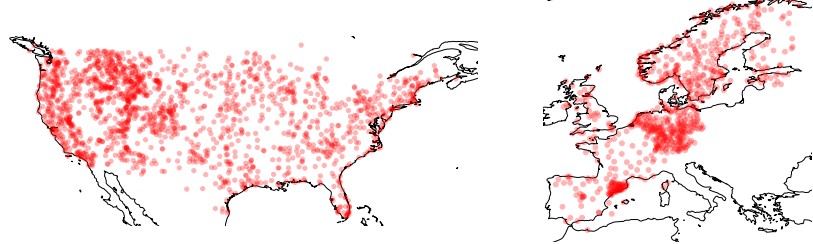

Figure 5: (left) GHCN weather stations in the USA and (right) GHCN weather stations in Europe.

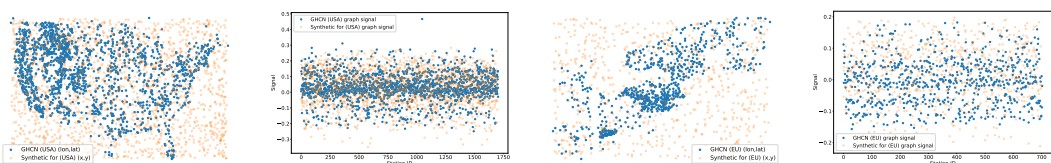

Figure 6: (first and third) Spatial distribution of the GHCN weather stations and synthetic nodes. (second and fourth) Function values of the GHCN records and synthetic function.

## B  TASK 2: GRAPH SIGNAL REGRESSION

### B.1  META-TRAIN

**Data**: For the graph signal regression task, we generate synthetic dynamics for meta-train tasks and the synthetic data is adapted to a target dataset. Before setting the topological configuration for the synthetic dynamics, we first examine the target dataset to understand its topological properties. Based on the number of stations and the scale of records, we tune the topological configuration for the synthetic dataset. We use $(N, F) = (1700, 2)$ for the USA records and $(N, F) = (700, 1.5)$ for Europe records, respectively, and 100 different initial values are generated to define different tasks. Fig. 5 visualizes how the regional stations are distributed and Fig. 6 demonstrates how the spatial distribution of synthetic nodes and scales of synthetic values are adapted to the corresponding target dynamics.

### B.2  META-TEST

**Data**: The GHCN-Daily summaries from land surface stations across the globe provide daily climate records from numerous sources. As the records from 100,000 stations in 180 countries and territories, the distribution of the weather stations is spatially non-uniform. We sample sensors from two different regions (1) the USA and (2) Europe and construct a graph structure from the regional stations based on $k$-NN algorithm ($k = 4$) as described in Defferrard et al. (2019). There are 1,705 and 703 fully functioning sensors in the USA and Europe, respectively. We use 2010 year records and first few daily records for few-shot training (5 and 10) and next 100/150 days for validation and test. Note that the number of learnable parameters is significantly reduced compared to those of the previous task to minimize overfitting as well as be comparable to other variants of graph neural networks.

Table 6: The number of learnable parameters for baselines and PiMetaL

|  | GCN | GAT | GraphSAGE | GN | PA-DGN | PiMetaL |
|---|---|---|---|---|---|---|
| # of parameters | 10,801 | 11,203 | 21,401 | 20,385 | 33,795 | 33,795 |

### B.3  EXPERIMENTAL DETAILS

#### B.3.1  BASELINES

Since the input length of the regression task is fixed (length=5), we can consider many variants of graph neural networks for the task. We concatenate the 5-step signals and feed it into Graph convolutional net-

Table 7: Regression error (MAE, $10^{-3}$) of different topology for synthetic dynamics (Europe)

|         | $(N, F) = (700, 1.5)$ | $(N, F) = (1700, 2)$ | $(N, F) = (128, 7)$ |
|---------|-----------------------|----------------------|---------------------|
| 5-shot  | **0.781±0.019**       | 0.981±0.131          | 1.007±0.096         |
| 10-shot | **0.773±0.014**       | 0.951±0.151          | 0.932±0.058         |

works (**GCN**) (Kipf & Welling, 2017), Graph attention networks (**GAT**) (Veličković et al., 2018), **Graph-SAGE** (Hamilton et al., 2017), and Graph networks (**GN**) (Battaglia et al., 2018) to predict next signals across all nodes. For the baselines, we commonly consider 3-hop neighbors of $i$-th node to predict of the $i$-th node and the number of learnable parameters is similar to provide similar expressive power.

## C    SENSITIVITY ANALYSIS OF SYNTHETIC DYNAMICS

It is important to study how much the model's performance is dependent on synthetic topology. In this section, we conduct an ablation study to see whether different choices of the synthetic topology affects the performance significantly or not. According to Table 4, the regression error is fairly converged from a few samples for the Europe records. Thus, we apply different synthetic topology for the data to see if the saturated regression error is significantly changed. For this ablation study, we reuse the synthetic dynamics adapted for the USA records and generate one more synthetic dynamics for spatially low-resolution cases.

Table 7 shows that the regression performance across different topology is stable regardless of the number of shots, however, it is significantly degraded when we change the synthetic topology from the adapted one ($(N, F) = (700, 1.5)$) for meta-training. When we increase the spatial resolution ($N = 700 \rightarrow 1700$), the meta-initialized spatial modules are adapted to learn spatial derivatives defined on spatially higher resolution. In such case, SDM likely assigns high weights to directly adjacent nodes as well as farther nodes (e.g., 3-hop nodes) as all neighbor nodes are strongly associated with exact spatial derivatives. On the other hand, if SDM is meta-initialized from a lower resolution ($N = 700 \rightarrow 128$), further nodes are too much underestimated. Thus, it is important to construct proper topology for transferring the PDE-independent representations from synthetic dynamics to target dynamics.

