# OpenReview forum: "Physics-aware Spatiotemporal Modules with Auxiliary Tasks for Meta-Learning"
_ICLR.cc/2021/Conference — Reject_

### Official Review · AnonReviewer2 · 2020-10-26
**A solid combination of PIML and multi-task learning.**

**Rating:** 8
**Confidence:** 3

**Review:**

The authors propose methodology for sharing learned differencing coefficients for estimating spatial derivatives between multiple spatio-temporal modeling tasks. They show that increased number of tasks improves learning. Additionally, the authors propose a meta-initialization procedure by which the differencing coefficients are initialized to values obtained from synthetic data. They show that this initialization procedure improves performance.

It seems to me that the 'share-ability' of the spatial differencing coefficients requires that each task is operating on the same spatial manifold with same or similar measurement locations. Toward that end, the initial differencing coefficients appear to be chosen using a flat 2D topology. The appropriateness of this choice is probably responsible for the improvements offered by the proposed approach. However, if the initial topology is poorly chosen, or the different tasks have significantly different topologies one would expect poor performance or negative transfer. How to identify these degenerate cases seems to be an open problem.

Aside from my rehashing the perennial question of 'when does task sharing help?' I found this paper to be well written, and a solid contribution to the field of physics-informed machine learning. Certainly, many applications include multiple different types of measurements at common locations which might be expected to transfer.

---

> ### Author Response · Authors · 2020-11-18
> **Author response to Reviewer 2**
>
> We thank the reviewer for your time assessing our work and the constructive feedback.
>
> ### When does task sharing help?
>
> > It is great intuition that the **share-ability** of spatial derivatives requires similar topological properties. In other words, if the generated synthetic dynamics are defined on very different topology compared to that of targeted dynamics, the PDE-independent/-invariant representations won’t be easily transferred to real-world topology from synthetic topology.
> In the updated manuscript (and the general response above), we included additional experiments to describe how the synthetic topology is defined based on that of the real-world dataset (Appendix B) and how the different synthetic topology affects the model’s performance **(Appendix C and the general response above)**.

---

### Official Review · AnonReviewer1 · 2020-10-27
**Interesting idea, clarifications needed**

**Rating:** 6
**Confidence:** 3

**Review:**

The paper describes the approach to meta-learning of spatiotemporal predictions for sparse data with auxiliary spatial derivative modules.

The paper provides novel insights on how to perform physically informed meta-learning and can be useful to the community, but the questions below need to be answered to assure that it is indeed meta-initialisation which improves the few-shot learning performance (the review rating reflects these questions, especially question 2 on the motivation of pre-training, and could be improved based on the answers):
1. It is said that *“While it has been shown that the data-driven approximation of spatial derivatives is more precise than that of finite difference method”*. At the same time, it is said that *'We perform two sets of experiments: evaluate few-shot learning performance (1) when SDM is trained from scratch; (2) when SDM is meta-initialized.’* The reviewer thinks that there might be a third scenario: while data-driven approximation might be more precise, the finite difference method provides for better features for PDE-agnostic methods. Has this scenario been accounted for or could it be used as an additional ablation study?

2. The pre-training has been performed on the synthetic data, and the authors mention the limitations of the approach: *“
Although introducing auxiliary tasks based on synthetic datasets improves the prediction performance, they need to be chosen and constructed manually and intuitively.”* In connection to that, there might be less scarce, non-extreme, events in the same dataset, which could be used for pretraining. For example, in the extreme weather dataset,  the top-10 extreme weather events since 1984 are selected; is it possible to use some of the remaining data for pretraining on non-extreme events? This could remove the bias of synthetic dataset hyper parameter tuning and could serve as  a baseline. Is it possible to say how would the method, pre-trained on such real non-extreme dataset, perform against the synthetic baseline in this case?

3. In Table 3, the experimental results are reported, with an (anticipated) tendency of diminishing return in spatial meta-initialisation as the number of T-shots increase; is there any way to see whether, as the value of T is increased, there will be the value T when pre-training does not improve or even has worse performance? When does it happen, and does it ever get worse than no-pretraining baseline?

---

> ### Author Response · Authors · 2020-11-18
> **Author response to Reviewer 1**
>
> We thank the reviewer for your time assessing our work and the constructive feedback.
>
> ### Finite difference method
>
> > It is a great suggestion to check if the finite difference method is enough to provide shareable representations.
> We conducted additional experiments for the finite difference method baseline **(FDM+RGN)**. Basically, the learnable spatial modules in the proposed method are replaced by finite modules, which numerically compute the spatial derivatives (gradients and Laplacian). Since we have graph signals as our main datasets, we defined the gradients as edge features $\nabla u_{ij}=\frac{u_j - u_i}{||x_j - x_i||}$ where $u_i$ is a graph signal at node $i$ and $x_i$ is a positional vector of the node $i$, and Laplacian as node features $\nabla^2 u_i=\sum_{j\in N_i}\nabla u_{ij}$.
>
> T-shot   | Method | AQI-CO | ExtremeWeather |
> :--------------------------:|:--------------------------:|:--------------------------:|:--------------------------:|
> 5-shot | FDM+RGN | 0.0291$\pm$0.0039 | 0.9883$\pm$0.5567 |
>  | PA-DGN | 0.0363$\pm$0.0090 | 0.9653$\pm$0.1384 |
>  | PiMetaL | **0.0253$\pm$0.0055** | **0.9167$\pm$0.0746** |
> 7-shot | FDM+RGN | 0.0258$\pm$0.0023 | 0.7626$\pm$0.0602 |
>  | PA-DGN | 0.0225$\pm$0.0018 | 0.7478$\pm$0.0199 |
>  | PiMetaL | **0.0182$\pm$0.0019** | **0.7274$\pm$0.0089** |
> 10-shot | FDM+RGN | 0.0213$\pm$0.0013 | 0.7090$\pm$0.0030 |
>  | PA-DGN | 0.0146$\pm$0.0005 | 0.4156$\pm$0.0145 |
>  | PiMetaL | **0.0115$\pm$0.0004** | **0.4066$\pm$0.0247** |
>
> > The finite difference method provides proxies of exact spatial derivatives and the representations are useful particularly when $T=5$ but its performance is rapidly saturated and it comes from the gap between the learnable spatial modules and fixed numerical coefficients. The results provide a new point of view on how to utilize synthetic or simulated datasets to handle challenges caused by limited datasets.
>
>
> ### Pre-training with same dataset
>
> > While it is an interesting idea to utilize non-extreme events in the same dataset for pre-training, this approach is not associated with our proposed method. First of all, we define **the auxiliary task as supervised learning for spatial derivatives**, and for the auxiliary task labels, **the exact spatial derivatives are only available in the simulated dynamics** since the simulated one is generated from an analytic equation. In the real-world dataset, especially graph signals which have a spatially low resolution, the exact spatial derivatives are not available and it is hard to be estimated due to the significant numerical error.
> In this work, we claim that **the spatial derivatives are PDE-independent/-invariant representations and they can be meta-learned from synthetic data** describing locally conserved quantities.
>
> ### The value T when pre-training does not improve or even has worse performance
>
> > For task 1 (multi-step graph signal generation), our datasets are extremely short across different tasks. Mostly, the length of each task is shorter than 20-steps, and thus, it is not possible to increase $T$ as much as we want. However, in the additional experiment (task 2, graph signal regression), we found that the performance is no more significantly improved when we increase the number of samples from 5 to 10 (See 5-shot (EU) and 10-shot (EU) in the general response above.). While the regression errors seem to be converged, our model provides a much lower error level than those of other baselines.

---

> > ### Comment · AnonReviewer1 · 2020-11-24
> > **Improvements in analysis**
> >
> > The reviewer thinks that the analysis has been significantly improved and recommends paper acceptance. The score has been raised accordingly. After reading general responses, the reviewer lists the reasons behind the score.
> >
> > Strengths:
> > - well-written paper
> > - novel methodology of physically informed meta-learning
> > - new experimental insights into the parameterisation of the auxiliary tasks
> > - experimental analysis is substantially improved
> >
> > Weaknesses:
> > - more evidence is needed on how to systematically address the question of constructing auxiliary tasks (but one might say that the purpose of the paper was to introduce this idea and to provide the methodology for physically informed meta-learning and this is not a topic of the paper)
> > - more is yet to be known about the limitations of the method and its performance in a range of settings (e.g. high-dimensional setting brought up by R4), but this is clearly described by the authors and mitigated by a range of new experiments in the rebuttal

---

> > > ### Author Response · Authors · 2020-11-25
> > > **Author response to R1**
> > >
> > > Thank you for your time reading and responding to our response.
> > >
> > > For future work, we think that there are several directions based on the reviews:
> > > - Automatically generating auxiliary tasks from real-world observations
> > > - Validating our proposed idea on the higher dimensional dynamics
> > > - Designing other auxiliary tasks based on not only spatial derivatives but also other physical properties

---

### Official Review · AnonReviewer4 · 2020-10-29
**This paper is well written. The idea of physics-awareness in meta learning seems novel and clear. However, some limitations in the approach remain to be improved.**

**Rating:** 5
**Confidence:** 4

**Review:**

This paper proposes a framework for physics-aware meta-learning to tackle the few-shot learning
challenges in physical observations. The authors claim that by incorporating PDE-independent
knowledge from simulated data, the framework provides reusable features to meta-test tasks with
a limited amount of data. The experiments on synthetic and real-world spatiotemporal prediction
tasks demonstrate the effectiveness of the framework. However, there still exist some limitations
to the approach.
Pros:
1. The idea of using meta-learning for modeling physics-related spatiotemporal dynamics is
novel.
2. The paper is well written, and the idea is presented clearly.
Cons:
1. How to choose and construct auxiliary tasks based on synthetic datasets is important for
the performance but there is no systematic approach to addressing such an issue.
2. The authors can add more sensitivity analysis to show how different choices of auxiliary
tasks can influence the experiment performance.
3. The paper focuses on modeling and predicting sensor-based observation in 2-dimension.
However, it is not clear if such an approach can be effective in 3 or higher dimension
space. Higher-dimension space and higher-order derivative can bring up challenges to
constructing useful auxiliary tasks.

---

> ### Author Response · Authors · 2020-11-18
> **Author response to Reviewer 4**
>
> We thank the reviewer for your time assessing our work and the constructive feedback.
>
> ### How to choose and construct auxiliary tasks?
>
> > We have provided this suggestion in **the general response above (Suggestion 2)**. For further information, we updated our manuscript **(Section 5.1 and Appendix B)**.
>
> ### Sensitivity of the different auxiliary tasks
>
> > We have provided this concern in **the general response above (Suggestion 3)**. We conducted an ablation study on how the regression performance is changed when the simulated dynamics are changed and the results are provided in the general response.
>
> ### High order
>
> > The reviewer brings-up a pretty interesting case.  If targeted dynamics are defined in 3D dimension space such as temperature across longitude/latitude/altitude, the simulated dynamics are supposed to be 3D as well. Since many variants of the continuity equation can be seamlessly defined in 3D or higher dimension (e.g., Navier-Stokes equation in 3D) and the exact spatial derivatives can be defined in all directions, we can easily extend our model to the 3D space by adding additional spatial modules for spatial derivatives along the z-axis. This constitutes our future work.

---

### Official Review · AnonReviewer3 · 2020-11-01
**The authors provide a framework for physics-aware meta-learning with auxiliary tasks.**

**Rating:** 6
**Confidence:** 2

**Review:**

The key contribution of the paper is not very known to reviewer. Is it employing the PDE-independent knowledge (spatial derivatives) from simulating the PDEs?
Can the authors highlight some of the key contributions of the paper?

---

> ### Author Response · Authors · 2020-11-12
> **Key contributions of the paper**
>
> Thanks for reviewing our paper and here is the summary of the key contributions of the work.
>
> ### Key contributions
>
> > While few-shot learning has been actively studied in image classification tasks, little has been done for such tasks: **modeling physical observations with few samples** such as climate measurements even though it is common that these types of data are limited in practice. One reason is that it is unknown what are the shareable/generalizable representations across different tasks for the physics-associated dynamics unlike for images.
> In this work, we first show that **spatial derivatives are associated with the shareable and invariant representations** across different physical dynamics inspired by a form of the continuity equation. Then, we propose **PDE-independent/-invariant spatial modules**, which are designed to learn the spatial representations via an auxiliary task, *supervised learning of spatial derivatives*. The spatial modules leverage simulated dynamics which can provide the invariant/shareable information across different dynamics. As the simulated dynamics can be easily generated, we can utilize large amounts of synthetic datasets for meta-training the spatial modules via auxiliary tasks, which are supervised learning for the spatial derivatives. Finally, we demonstrate **the effectiveness of the meta-trained spatial modules** on the graph signal prediction tasks of real-world dynamics with few observations.
> To the best of our knowledge, we are the first to provide PDE-independent/-invariant representations that are reusable across different physics-related dynamics and use the physics-inspired inductive bias under the meta-learning settings to tackle the limited data issue which is pretty common for real-world data.

---

### Official Review · AnonReviewer5 · 2020-11-07

**Rating:** 5
**Confidence:** 3

**Review:**

 The paper proposes to modular PDE solutions in networks into spatial and temporal modules, where spatial modules are rapidly adapted by meta learning.

Strengths

This paper is clear and relatively easy to read.

The paper does a reasonable of citing related work and the approach seems novel  (although incremental) to my knowledge.


Weaknesses

The overall quantitative benefits of the approach appear to be low. For example in table 2 or table 3, many of confidence intervals overlap. Furthermore, these results are shown in the best setting of the model -- when nearby tasks are drawn almost directly from the same training distribution. Furthermore

The overall technical novelty of approach seems to be low. Modularity with meta learning has been explored before, and the physical learning framework has also been learned before. This work seeks to combine both but does not seem have to have impressive performance.

Given that a central claim in the paper is that "We extract shareable parameters in the spatial
modules from synthetic data, which can be generated from different dynamics easily." I think it would be good to empirically test this result and  synthetic data is generated more diversely. For example, in Table 1/2, what happens when the underlying uis have different functional forms? What if only the test task has a separate functional form?

There are only three evaluated datasets in the entire paper -- one being a toy problem to visualize the impact of the approach. Since PDEs occur in so many different disciplines, it would be good to evaluate on additional datasets to more throughly validate the utility of this approach. For example, it may also be interesting to test this on additional non graph based settings.

### Post Rebuttal Update

I thank the authors for their response. However, I still have some remaining concerns about novelty, since to me this paper reads as another application of MAML to domain X, to speed up performance in the particular domain. Furthermore, I think it would be good to evaluate generalization to larger extent, for example on different conformations of dynamics as opposed to a fixed set of parameters on the synthetic dataset.

---

> ### Author Response · Authors · 2020-11-18
> **Author response to Reviewer 5**
>
> We thank the reviewer for your time assessing our work and the constructive feedback.
>
> ### The overall quantitative benefits are too low. The overall technical novelty seems to be low
>
> > While it is common that we only have access to a limited amount of data in physical systems in the real world, there have been few attempts to address this problem. Meta-learning has been introduced to handle few-shot learning efficiently, however, there are challenges for the existing meta-learning methods for modeling natural phenomena because it is not clearly known **(1) which representations need to be extracted** and **(2) how to extract the representations for different spatiotemporal observations in the real world**.
> Thus, it is very hard to define a shareable feature extractor for the physics-associated dynamics and prepare a large amount of data to pretrain the extractor to learn generalizable representations that are easily adaptable to unseen few-shot samples.
>
> > This work is one of **the first to explore effective representations shareable/generalizable across different physics-associated dynamics without knowing exact forms of PDEs** and **how the PDE-independent/-invariant representations can be extracted from simulated datasets to tackle limited data challenge** in real-world applications. To the best of our knowledge, there are very limited works about how to leverage simulated dynamics for efficient learning of data-driven models with few physics-guided observations and we think that this work provides a research direction for physics-guided/-informed deep learning.
>
> > To support the proposed method more thoroughly, we conducted **additional experiments** and provided the results in the updated manuscript (**Section 5.3**) as well as **the general response to all reviewers above**. From the results, we can see that the proposed method achieved significant improvement in reducing regression error by utilizing simulated dynamics to meta-train spatial modules, which are designed to learn PDE-independent/-invariant representations.
>
> ### Regarding different dynamics for synthetic dataset
>
> > It is important to generate diverse synthetic data to meta-train the spatial modules. For each configuration (N,E,F), we have generated 100 different sequences and each sequence has different initial values, which are randomly generated. (See Fig. 2 in the draft) In other words, the generated sequences are fairly different even if they have the same configuration due to the random initial values and thus, the experimental results (Table 2) cover the diverse enough simulated dynamics.
>
> ### What if only the test task has a separate functional form?
>
> > This question is directly addressed on the experimental results on the real-world dynamics since we don't have a closed-form PDE for the dynamics explicitly. In other words, while the test task's functional form is unknown, we successfully show that the meta-initialized spatial modules by simulated PDEs are effectively adapted to the different functional forms. We also added additional experimental results to validate the proposed idea more thoroughly. Please check **the general response (Suggestion 1 and 3) and the updated manuscript (Section 5.3 and Appendix C) for more details**.
>
> ### Limited datasets
>
> > Thanks for the suggestion. We have conducted an additional experiment (Graph signal regression) described in DeepSphere (Defferrard et al.) to thoroughly validate the proposed idea and to analyze how different auxiliary tasks influence the performance. Please check **the general response (Suggestion 1) and the updated manuscript (Section 5.3)** for more details.

---

### Author Response · Authors · 2020-11-18
**A general response to all reviewers**

We thank all reviewers for their valuable comments, which help us to improve the quality of this paper. Here are several questions/suggestions shared by all reviewers, therefore we provide a general response as follows:

### Suggestion 1: Evaluate on additional datasets to validate the utility of this approach.

Following this suggestion (R5: There are only three evaluated datasets), we have conducted an additional experiment (Graph signal regression) described in DeepSphere (Defferrard et al.) to thoroughly validate the proposed idea and to analyze how different auxiliary tasks influence the performance.

**Dataset**

> For the graph signal regression task, we consider **Global Historical Climatology Network (GHCN)** datasets provided by National Oceanic and Atmospheric Administration (NOAA) (https://www.ncdc.noaa.gov/ghcn-daily-description). Specifically, we split the GHCN data spatially into two regions: (1) the USA (1,705 stations) and (2) Europe (EU) (703 stations).

**Task**
> We consider a regression task where we predict the temperature $x_t$ from the temperature on the previous 5 days ($x_{t-5}:x_{t-1}$).

**Results**
These results are presented in Table 4 in the updated manuscript.

T-shot (region)&nbsp;&nbsp;    | GCN | GAT | GraphSAGE | GN | PA-DGN | PiMetaL
:--------:|:--------:|:--------:|:--------:|:--------:|:--------:|:--------:|
5-shot (USA) | 2.742$\pm$0.120&nbsp;&nbsp; | 2.549$\pm$0.115&nbsp;&nbsp; | 2.128$\pm$0.146&nbsp;&nbsp; | 2.252$\pm$0.131&nbsp;&nbsp; | 1.950$\pm$0.152&nbsp;&nbsp; | **1.794**$\pm$**0.130**
10-shot (USA) | 2.371$\pm$0.095&nbsp;&nbsp; | 2.178$\pm$0.066&nbsp;&nbsp; | 1.848$\pm$0.206&nbsp;&nbsp; | 1.949$\pm$0.115&nbsp;&nbsp; | 1.687$\pm$0.104&nbsp;&nbsp; | **1.567**$\pm$**0.103**
5-shot (EU) | 1.218$\pm$0.218&nbsp;&nbsp; | 1.161$\pm$0.234&nbsp;&nbsp; | 1.165$\pm$0.248&nbsp;&nbsp; | 1.181$\pm$0.210&nbsp;&nbsp; | 0.914$\pm$0.167&nbsp;&nbsp; | **0.781**$\pm$**0.019**
10-shot (EU) | 1.186$\pm$0.076&nbsp;&nbsp; | 1.142$\pm$0.070&nbsp;&nbsp; | 1.044$\pm$0.210&nbsp;&nbsp; | 1.116$\pm$0.147&nbsp;&nbsp; | 0.831$\pm$0.058&nbsp;&nbsp; | **0.773**$\pm$**0.014**

> The table above shows the results of the graph signal regression task across different baselines and the proposed method. There are two patterns in the results. First, although in general we observe an improvement in performance for all methods when we move from the 5-shot setting to the 10-shot setting, PiMetaL’s performance yields the smallest error. For the EU dataset, while 5-shot seems enough to achieve stable performance, it demonstrates that the PDE-independent representations make the regression error converge to a lower level. Overall, the experimental results prove that the learned spatial representations from simulated dynamics are beneficial for learning on limited data.
We have updated the main draft to include the additional experimental results **(See Section 5.3)** for further details.


### Suggestion 2: How to choose and construct auxiliary tasks (or synthetic datasets)?

> Indeed the choice of the synthetic data does depend on the specific application. Throughout the paper, we used the convection-diffusion equations to generate (spatially continuous) simulated dynamics (Long et al. PDE-Net) and sampled a finite number of nodes to construct graph signals.
When we generate simulated dynamics, sample the nodes, and construct graph structure, there are 3 parameters we can tune.
N: The number of sampled nodes
E: The number of adjacent nodes in a graph structure
F: The spatial frequency (it determines the function’s fluctuation and scale)
For **the task 1 (Section 5.2 Multi-step graph signal generation)**, we generated 100 different sequences (sequence length 20) per one combination of (N,E,F) where N$\in${256,625}, E$\in${4,8}, and F$\in${2,5}. These parameters were selected to cover the topological properties of the target datasets. For AQI and ExtremeWeather datasets, the number of sensors (nodes) is varying from 40 to 200, the number of adjacent nodes is 4 and 6, respectively. First, we intended to generate finer resolution (more nodes) synthetic dynamics than that of the target dataset to train SDM with the finer resolution data and apply it to coarse resolution data. Second, we set E and F to encompass the topological properties of the target data.

> For **the additional task 2 (Section 5.3 Graph signal regression)**, we generated synthetic dynamics for meta-train tasks and the synthetic data is adapted to a target dataset. Before setting the topological configuration for the synthetic dynamics, we first looked into the target dataset to understand its topological properties. Based on the number of stations and the scale of records, we tuned the topological configuration for the synthetic dataset. We used (N,F)=(1700,2) for the USA records (1705 nodes) and (N,F)=(700,1.5) for Europe records (703 nodes), respectively, and 100 different initial values were generated to define different tasks.

---

> ### Author Response · Authors · 2020-11-18
> **A general response to all reviewers (contd.)**
>
> ### Suggestion 3: Sensitivity analysis of synthetic dynamics
>
> > To study the importance of appropriate synthetic data choice, we performed an ablation study on whether different choices of the synthetic graph structure affects the performance significantly or not. According to the  Table above (Table 7 in the updated manuscript), the regression error converges under the few-shot setting for the Europe records. We applied different synthetic topology for the Europe records to see if the regression error significantly changes. For this ablation study, we reused the synthetic dynamics adapted for the USA records and generated one more synthetic dynamics for spatially low-resolution cases.
>
> | | $(N,F)=(700,1.5)$ | $(N,F)=(1700,2)$ | $(N,F)=(128,7)$ |
> |:--------------------------:|:--------------------------:|:--------------------------:|:--------------------------:|
> |5-shot | **0.781$\pm$0.019** | 0.981$\pm$0.131 | 1.007$\pm$0.096 |
> |10-shot | **0.773$\pm$0.014** | 0.951$\pm$0.151 | 0.932$\pm$0.058 |
>
> > The results show that the regression performance across different topology is stable regardless of the number of shots, however, it significantly degrades when we change the synthetic topology from the adapted one ((N,F)=(700,1.5)) for meta-training.
> When we **increase the spatial resolution (N=700$\rightarrow$1700)**, the meta-initialized spatial modules are adapted to learn spatial derivatives defined on spatially higher resolution.  In such a case, SDM likely assigns high weights to directly adjacent nodes as well as farther nodes (e.g., 3-hop nodes) as all neighbor nodes are strongly associated with exact spatial derivatives.
> On the other hand, if SDM is meta-initialized from **lower resolution (N=700$\rightarrow$128)**, farther nodes are severely underestimated.

---

### Author Response · Authors · 2020-11-19
**Updated revision**

We uploaded an improved manuscript thanks to the reviewers' comments.

The main update (green colored in the manuscript) is as follows:

* **Section 5.3: Graph signal regression**

  - We added additional experimental results. The experimental setting/results are briefly described in **A general response (Suggestion 1)** below and further details are in **Section 5.3 and Appendix B** in the updated revision.

* **Appendix C: Sensitivity analysis of synthetic dynamics**
  - We performed an ablation study on whether different choices of the synthetic graph structure affect the performance significantly or not. **A general response (Suggestion 3)** and **Appendix C** provide details.

* **Section 5.2: Finite difference method**
  - We conducted additional experiments for the finite difference method baseline (FDM+RGN). Basically, the learnable spatial modules in the proposed method are replaced by finite modules, which numerically compute the spatial derivatives (gradients and Laplacian). We provide the results in **Table 3** in the updated manuscript and **Author response to Reviewer 1** below.

A link to a public git repository containing all the code will be added after the blind-review process.

---

### Comment · Area_Chair1 · 2020-11-23
**Author Response**

Dear reviewers:

We are about to end the second discussion stage. Could you please check whether the authors have addressed your concerns and questions and potentially ask any further clarification questions?

Thank you, Your Area Chair

---

### Decision · Program_Chairs · 2021-01-07
**Final Decision**

**Decision:**

Reject

**Comment:**

This paper was reviewed by 5 experts in the field. The reviewers raised their concerns on lack of novelty, unconvincing experiment, and the presentation of this paper, While the paper clearly has merit, the decision is not to recommend acceptance. The authors are encouraged to consider the reviewers' comments when revising the paper for submission elsewhere.